# Effect of Martensite Volume Fraction on Oxidative and Adhesive Wear

**DOI:** 10.3390/ma14112964

**Published:** 2021-05-31

**Authors:** Yunbo Zhang, Abdeljalil Jourani

**Affiliations:** FRE UTC-CNRS 2012 Roberval, Université de Technologie de Compiègne, Alliance Sorbonne Université, Centre de Recherche Royallieu, CS 60 319, CEDEX, 60 203 Compiègne, France; yunbo.zhang@utc.fr

**Keywords:** microstructure, dual phases, friction, oxidative wear, adhesive wear

## Abstract

It is generally known that microstructure can considerably affect the tribological behavior of non-lubricated rubbing. However, there is still a lack of awareness about the effect of microstructure on oxidative wear. The present study focused on the effect of martensite volume fraction (MVF) on oxidative wear by using 25CD4 dual-phase steel. Dry friction tests were performed on a ball-on-flat tribometer with a normal load of 15 N and a mean sliding velocity of 0.013 m/s. Friction coefficient and wear rate increase with the increasing MVF. SEM observation and EDXS analyses of the wear scars showed that the oxidation increases with decreasing MVF. For lower MVF, the main wear mechanism is mild oxidative wear. For higher MVF, severe adhesion is predominant as a wear mechanism. The size of the debris decreases with decreasing MVF.

## 1. Introduction

Wear is one of the main reasons for the failure of machine elements (wear, corrosion, fracture and fatigue) [1]. In addition, material wear greatly affects the service life of mechanical components. It may cause the failure of mechanical parts which then leads to a great loss in energy and materials, eventually associated with a consequent financial cost. Therefore, studying the friction and wear of materials is fundamental to reduce the severe wear of mechanical parts, as well as to reduce energy and economic losses.

In the last few years, many researchers have contributed to the study of the impact of microstructure on friction and wear mechanisms [2,3,4,5,6,7]. For dual-phase steel, Davie et al. [8] suggest that the strength of the microstructure is related to the grain size of ferrite and the MVF, and irrelevant to the composition and strength of the martensite. Some investigations are concentrated on the effect of MVF on tribological behavior [9,10,11,12,13,14,15]. Trevisiol et al. [13] show that the friction coefficient and wear rate decrease with increasing MVF. For low MVF, the main wear mechanisms are plowing and adhesive wear, while cutting wear becomes predominant instead of plowing and adhesion under high MVF. Saghafian and Kheirandish [12] explain that the wear rate of dual-phase steels decreases with increasing MVF, which is probably due to the reduction of the real contact area and the formation of cracks. However, Jha et al. [9] suggest that for dual-phase steel, compared with the full martensitic microstructure, better wear resistance can be provided by a microstructure with a small content of soft ferrite. To a certain extent, although the presence of ferrite reduces the overall strength and hardness of the material, it reduces the brittleness, enhancing the overall fatigue resistance and wear resistance. In addition, scratch tests on dual-phase steels at low normal load show that the wear depth decreases with increasing MVF [14,15]. On the other hand, under high normal load, with increasing MVF, it first decreases and then increases. However, the optimal MVF which displays the best scratch resistance is yet to be determined. Hence, further studies are needed to investigate the effect of MVF on friction and wear. When considering the wear mechanism, there are fewer investigations about the impact of microstructure such as phase volume fractions on oxidative wear. Previous research of the microstructural factors demonstrates that the wear rate of the dual-phase steel decreases linearly with an increasing MVF for a particular load under a pin-on-disc machine. The main reason for the decline in the coefficient of friction is the decreasing actual contact area caused by the increasing material hardness. Similarly, the average coefficient of friction decreased with an increasing MVF at a given load [16,17,18]. The above studies have all involved the impact of MVF on the wear rate of dual-phase steel, but there are huge differences in the experimental conditions of friction. The method adopted by Saghfian et al. [12] was a unidirectional sliding contact of a cylindrical steel pin the steel disc, with a load of 61–83 N, and a relatively high sliding speed (1.2 m/s). The experiments of Trevisiol et al. [13] were focused on the contact of cylindrical pin samples against the abrasive papers. The normal load was also relatively high (50~110 N), and the sliding speed was constant at 0.06 m/s. The test of Jha et al. [9] was performed by a rubber wheel tribometer, in which the rubber wheel was directly rubbed against the flat steel sample in the vertical direction. The experimental load reached 48 N and the sliding velocity was 3.27 m/s. Due to the different tribological systems, the wear mechanism of dual-phase steel is constantly changing. For example, for the frictional system of Trevisiol et al. [13], the main wear mechanism is abrasive wear, while in the tribological system of Saghafian et al. [12], the main wear mechanism is delamination wear. Considering that different tribological systems have a great effect on the properties of friction and wear, it is necessary to establish different frictional systems to investigate the relationship between their corresponding tribological behaviors and microstructures. Therefore, using the friction condition established in this paper to investigate the effect of the microstructure of dual-phase steel on oxidative wear is a good supplement to the investigations of the tribological behavior of dual-phase steel.

The present work is to investigate the effect of microstructure on oxidative wear of dual-phase steel under low surface roughness. Dry friction tests have been performed by using a ball-on-flat tribometer in air. Wear rate was obtained and calculated using a 3D optical profilometer. Wear observation and wear mechanism analysis were provided by scanning electron microscopy (SEM) and energy-dispersive X-ray spectroscopy (EDXS).

## 2. Materials and Methods

### 2.1. Material and Heat Treatments

The friction tests were carried out by using 25CD4 steel, the chemical composition of which is shown in Table 1. The diameter and the thickness of each sample were 55 mm and 3 mm, respectively.

A step quench heat treatment was carried out for the specimens to acquire the dual-phase microstructures with martensite and ferrite. As revealed in Figure 1, this proceeds in two stages: for the first step, a full austenitization was performed at 900 °C for 30 min; then, to attain the dual-phase region consisting of austenite and ferrite, the specimens were kept at an intercritical temperature of 725 °C. At 725 °C, the established microstructures comprised austenite and ferrite which was transformed from austenite. Lastly, the specimens were subjected to a water quench to allow the transformation from austenite to martensite while ferrite remained unchanged. In the process of the intercritical heat treatment at 725 °C, all samples were kept under different holding times, changing from 0 min to 7 min, to obtain various MVFs.

### 2.2. Microstructure Characterization

Before the friction tests, the six specimens with unequal MVFs were ground by abrasive papers with an abrasive particle grit of 80. The aim of this process was to remove the oxide layer caused by the heat treatment and to attain the structure of the matrix material. After that, the mechanical grinding was carried out with abrasive paper with a grit size ranging from 180 to 4000. Next, the surfaces were polished separately with diamond pastes of 3 µm and 1 µm. Then, utilizing ultrasound, the polished surfaces were cleaned in ethanol for 10 min. After this step, the specimens were etched with 1% Nital solution for three seconds and immediately cleaned by ultrasonic vibration for 10 min. To obtain the MVF of every sample, photographs of microstructures were taken by a scanning electron microscope ZEISS Sigma (ZEISS, Oberkochen, Germany) at 20 keV. The image processing was applied through ImageJ to determine the volume fraction of martensite and ferrite.

The measurements of macrohardness were completed by a Vickers indenter under a normal load of 10 kg. The value of macrohardness is represented by the average value of ten measurements obtained from different positions.

The measurements of microhardness were carried out utilizing a ZHV-µs (H115) Micro Vickers Hardness Tester (Zwick, West Midlands, UK) under a load of 0.01 kg. The microhardness value was determined by the average of three measurements. For each measurement, two positions were chosen at the edge and one position was chosen in the center at each phase.

### 2.3. Friction Tests

The frictional experiments were carried out with a ball-on-flat testing configuration on a reciprocating tribometer (CSM, Peseux, Switzerland), which is shown in Figure 2. The ball with a radius of 3 mm was made from ceramic Al_2_O_3_ and was commercially available as a standard sample. The surface roughness was between 0.25 µm and 0.35 µm for the disc samples. The specimens were tested at normal loads of 15 N and at a mean sliding velocity of 0.013 m/s. This low sliding velocity was applied to prevent high frictional heat on the contact surface during the friction process. The entire sliding distance was 7 m with a stroke length of 10 mm. The friction coefficient could be obtained directly by the tribometer through the frictional force sensor. Before the friction tests, each sample was cleaned for 10 min by ultrasound in ethanol, and then the surface was kept dry. All experiments were carried out at room temperature (23 °C) in an air atmosphere, and the relative humidity was maintained at 50–60%. Each test in a given condition was repeated three times with the same samples in different locations to ensure the reliability of the test results.

The topography of the worn surface was analyzed using a 3D optical profilometer (Sensofar Metrology, Barcelona, Spain) to measure the wear rate. Methods and steps are shown in Figure 3. For each scar, three different locations were selected as measurement samples using the SensoSCAN (Sensofar, Barcelona, Spain). Similarly, profile measurements for each surface sample were completed three times to acquire the area of the peak and hole by the SensoMAP application [19]. Finally, the average of the nine measurements was used to represent the final wear accumulation of the wear track.

It can be seen from the cross-sectional profile of the wear track (Figure 3) that the ‘peak’ refers to plastic deformation due to extrusion during the friction process, and the ‘hole’ refers to the wear loss of the disc material during the friction process. Therefore, the wear rate *Q* was calculated:(1)$$Q=\frac{\left(S_{hole}-S_{peak}\right)\times L}{F\times D}$$
where $S_{hole}$ and $S_{peak}$ are the areas of the hole and peak, L is the length of the scar, F means the normal load and D means the total sliding distance.

The final wear rate under the same experimental parameter was determined by the average of at least three tests.

After tribological tests, a scanning electron microscope (ZEISS Sigma) was applied to observe and analyze the wear tracks. In addition, EDXS analysis was performed for the wear debris and scars to investigate the wear mechanisms.

## 3. Results and Discussion

### 3.1. Microstructure and Hardness Analyses

It is well accepted that at a given critical annealing temperature, the volume fractions of ferrite and martensite are determined by the holding time [20,21]. In Figure 4, the SEM pictures show the microstructures composed of martensite and ferrite at varied holding times. First, during the austenitization, a complete austenitic microstructure is produced. Then, as the temperature decreased from 900 °C to 725 °C, ferrite nucleates at the austenite grain boundaries. Finally, the austenite transforms to martensite by a water quenching, resulting in little retained austenite, while the ferrite formed at the critical temperature remains unchanged. It is well accepted that retained austenite depends on the carbon content of the steel. For steels with carbon below 0.5%, the residual austenite is below 2% [22]. Therefore, for 25CD4, only martensite and ferrite can be observed in the micrographs but little retained austenite (less than 2%) is distinguished [23].

The curve of the MVF varied with holding time at 725 °C, and is shown in Figure 5. The result shows that the MVF decreases with the increasing intercritical holding time. The specimen with a holding time of 0 min, and water quenched directly after austenitizing, exhibits the highest MVF of 100%. The MVF then decreases with increasing holding time. Finally, the sample with a holding time of 7 min presents a minimum MVF of 48%. At the critical temperature, due to the increased holding time, the ferrite formed due to the austenite transformation increases.

The macrohardness as a function of the MVF is shown in Figure 6. The specimen with the maximum macrohardness is the one directly water quenched after austenitizing, and its macrohardness reaches over 600 HV. As the MVF decreases, the macrohardness of the specimens progressively declines. When the holding time is 7 min, the MVF is 48%, and the minimum value of macrohardness of about 300 HV is obtained. As noticed in many previous studies [10,24,25], the macrohardness increases owing to the increasing hard MVF and decreasing soft ferrite volume fraction. As we all know, the hardness of the phase is determined by the carbon content. The martensite structure is a metastable iron phase, and its carbon content is much larger than that of solid solution. Due to the diffusionless transformation from austenite into martensite, martensite is given a high hardness. In addition, the maximum carbon content dissolved in ferrite is only 0.02%, which makes the hardness of ferrite lower than martensite. Some works have shown [17,23,26] that even if the hardness of the individual phase follows the opposite trend, the overall macrohardness of dual-phase steel increases with the MVF. In addition, as described in some work [27,28], the carbon content in the separated martensite increases with the decreasing MVF, resulting in increasing microhardness.

The carbon content of the dual-phase steel can be acquired by the equation [13]:(2)$$C=C_{f}V_{f}+C_{m}V_{m}=C_{f}\left(1-V_{m}\right)+C_{m}V_{m}$$

Further, the carbon content of martensite can be shown as:(3)$$C_{m}=C_{f}+\left(C-C_{f}\right)/V_{m}$$

Among them, *C_m_* is the carbon concentration of isolated martensite, *C_f_* is the carbon concentration of ferrite, C is the carbon content of dual-phase steel, *V_m_* and *V_f_* are the volume fraction of martensite and ferrite, respectively. Based on the above expression, the carbon content of martensite decreases with increasing MVF. There is a high carbon content in the martensitic phase under a lower MVF. For a high MVF, the content of carbon is relatively low.

As shown in Figure 7, the results of microhardness experiments also confirm the above conclusion. In addition, the measurement points selected in the microhardness experiment are at the edge and center of each phase (Figure 8). Further experimental results show that for martensite, the microhardness in the center is greater than near the edge. Due to the lower amount of dissolved carbon in the ferrite, the carbon at the austenite grain boundary is transferred from the edge to the center during the nucleation of ferrite in the austenite grain boundary, which is more beneficial for ferrite nucleation and growth. After water quenching, because of the transformation from austenite to martensite, the carbon content in the center of the formed martensite is higher than at the edge, so the microhardness of the center is greater than that of the edge. For ferrite, since the quantity of carbon dissolved in the ferrite is low, there is no significant difference in microhardness between the center and the edge of ferrite.

### 3.2. Friction Coefficient

Figure 9 shows the variation of the friction coefficient with MVF. The average value of the friction coefficient at a steady state is acquired and calculated by the experimental data from 100 to 546 s. Figure 10 shows the average value of the friction coefficient at a steady state as a function of MVF and it remains 0.49–0.60. It is worth noticing that the minimum friction coefficient appears on the sample with an MVF of 48%, with a value of 0.49. The maximum friction coefficient also appeared on the specimen with an MVF of 100%, and its value reached 0.60. As can be seen in Figure 9 and Figure 10, the coefficient of friction declines with decreasing MVF.

### 3.3. Wear Rate

For wear, the cumulative wear can be obtained by the method described previously (Figure 3). The trend of wear rate with relation to MVF is shown in Figure 11. There is low wear on the sample with low MVF (59% or less) compared with high MVF (93% or more). In addition, it can be seen that the minimum wear occurs on the sample with the MVF of 59% and the maximum occurs on the sample with 93% MVF. Meanwhile, for the samples with MVF between 59% and 93%, the wear rate increases with the increasing MVF.

### 3.4. Wear Mechanisms

Figure 12 shows SEM images of the wear tracks under different MVFs. It can be seen that with the MVF decreasing, the wear debris seems to be well-adhered to the wear track, and even formed a layer of compressed thin film which smeared across the wear track.

Table 2 shows the oxygen concentration of the wear track surface obtained with EDXS analysis. The oxygen concentration of samples with low MVF is much higher than that with high MVF samples. This indicates that the surface of the dual-phase steel produces more oxidative wear as MVF decreases. It is well known that the formation of iron oxides (such as Fe_2_O_3_ and Fe_3_O_4_) [29,30,31], compared with non-oxidized metal surfaces, reduces the adhesion. This provides the basis for the observation regarding the adhesive wear. The results of EDXS analysis on the points of the wear track surface are shown in Figure 13 and Table 3. It can be seen that the oxygen content of the wear debris was high, which also indicates that significant oxidative wear occurred during the experiment. It can be found that the dark or black areas of the wear track have higher oxygen concentrations, while the light gray areas have low oxygen concentration (3%). From the EDXS results and SEM micrograph analysis, a great number of tribo-oxides appeared on worn surfaces of dual-phase steel as MVF decreased. The morphology of worn surfaces gradually presented smooth un-delaminated regions which were typical tracks of oxidative wear. Therefore, the previously mentioned compression film formed by wear debris can be determined as an oxide film. Due to the existence of the oxide film, it effectively prevents direct contact with the frictional matrix. As the friction continues, the debris of oxides that remains in the wear track is better compacted on the contact surface. This also promotes the formation and maintenance of the oxide layer. Due to the existence of the hard oxide layer, the wear rate gradually decreases, especially the adhesive wear, which was also observed in this experiment.

Figure 14 shows the evolution of abrasive particle size as a function of MVF. It can be concluded that the size of debris decreases with the MVF.

Combining SEM images and EDXS analysis results, the tribological behaviors of dual-phase steel are analyzed and discussed as the MVF varies.

As we all know, metal oxides generally provide relatively favorable friction properties [32,33], and direct contact of metallic Fe causes severe adhesion [29,31]. The higher adhesion caused by metal-to-metal contact dominates the tribological behavior of the contact points. Previous studies have shown that the increased friction coefficient is caused by dominant adhesive wear [13]. In addition, the delamination of the compacted transfer layer and the formation of oxide can also be a foundation. Compared with adhesive wear, the compacted oxide layer can effectively reduce the real contact area and junctions, which may bring down the friction coefficient. The results of EDXS analysis show that the oxidative wear increases with decreasing MVF, which explains the phenomenon that friction coefficient decreases with decreasing MVF.

The main wear of samples with low MVF is oxidative wear because of the relatively high oxygen concentration of the wear track. The existence of a composite oxide film is like a protective layer on the metal surface [34,35,36,37,38], which reasonably explains why the wear of low MVF is lower than that of high MVF.

Figure 15 shows the acquisition method and calculation principle of the plastic deformation in this experiment. In addition, Figure 16 shows the evolution of plastic deformation in relation to MVF, which also provides a supplement to explain the variation in friction coefficient and wear rate in these experiments.

The increase in plastic deformation represents an increase in oxidative wear. Under the conditions of mild oxidative wear theory, the increase in oxidative wear means a decrease in wear rate.

During the sliding process, adhesion and junctions occur only on the real contact areas, where high contact pressure causes the plastic deformation of the matrix. Compared with other regions, there is a higher temperature at plastically deformed contacting areas. Thus, these plastically deformed areas are preferential locations for oxidation [16,35]. As the plastic deformation of the substrate increases, more cracks would be formed and propagated [39,40]. This could easily lead to oxygen entering these cracks and reacting with the steel substrate to form oxides inside the matrix. A large plastic deformation would occur in a softer matrix under the same normal load and sliding speed, and so produce more tribo-oxides. Due to the low normal load and sliding velocity, the plastic deformation generated during wear is helpful to retain more tribo-oxides. The model proposed by Quinn [41,42] can also be used to show that mild oxidative wear is usually generated under low normal load and velocity, and wear property is only related to tribo-oxides. As the low-speed friction proceeds, the oxide remaining in the wear track is compacted into a hard tribo-oxide layer to prevent metal–metal contact and thus reduce wear.

According to the results of the macrohardness test (Figure 6) and the plastic deformation measurement (Figure 16), it can be determined that the macrohardness of the dual-phase steel matrix decreases with decreasing MVF, and the plastic deformation increases with decreasing MVF. Meanwhile, owing to the oxide increases during plastic deformation, oxidative wear increases with decreasing MVF. This reasonably explains the relationship between the microstructure of dual-phase steel and oxidative wear.

## 4. Conclusions

The current research was concentrated on determining the friction and wear mechanism of the tribological behavior of 25CD4 dual-phase steel with different MVF contacts in ball-on-flat conditions (non-lubricated sliding under a normal load of 15 N). In the selected operating conditions, the main conclusions can be drawn:(1)The friction coefficient decreases with decreasing MVF;(2)For samples with low MVF (59% or less), there is little wear compared with high MVF;(3)As the MVF decreases, the oxidative wear increases. The formation of a compact oxide film (glazed layer) can effectively reduce wear, especially adhesion wear. For low MVF, compact oxide layers are more likely to form and remain on worn surfaces;(4)For lower MVF, the main wear mechanism is mild oxidative wear. However, for higher MVF, severe adhesion wear is predominant;(5)The plastic deformation decreases with increasing MVF;(6)The size of the debris decreases with the MVF.

This work is more applicable to the tribological behavior of materials in a stable state. There is a lack of knowledge about the friction and wear behavior in the initial running-in state. In addition, due to the limitations of equipment and technology, the experimental results were mainly analyzed and interpreted through surface characterization. In future work, more investigations may explore the perspective of grain structure and energy changes.

## Figures and Tables

**Figure 1 materials-14-02964-f001:**
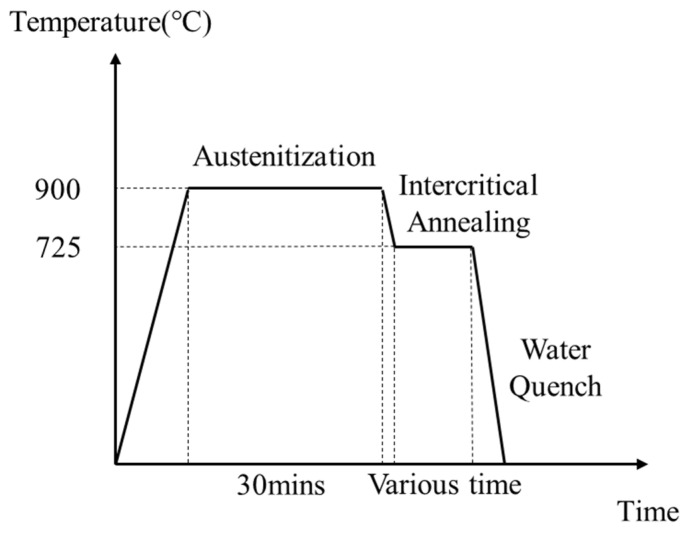
Schematic drawing of the step quenched heat treatment for specimens.

**Figure 2 materials-14-02964-f002:**
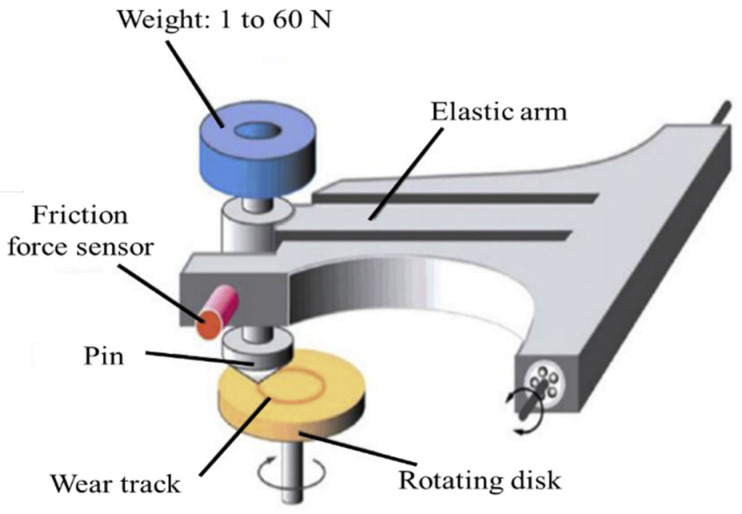
Tribological test setup and principle.

**Figure 3 materials-14-02964-f003:**
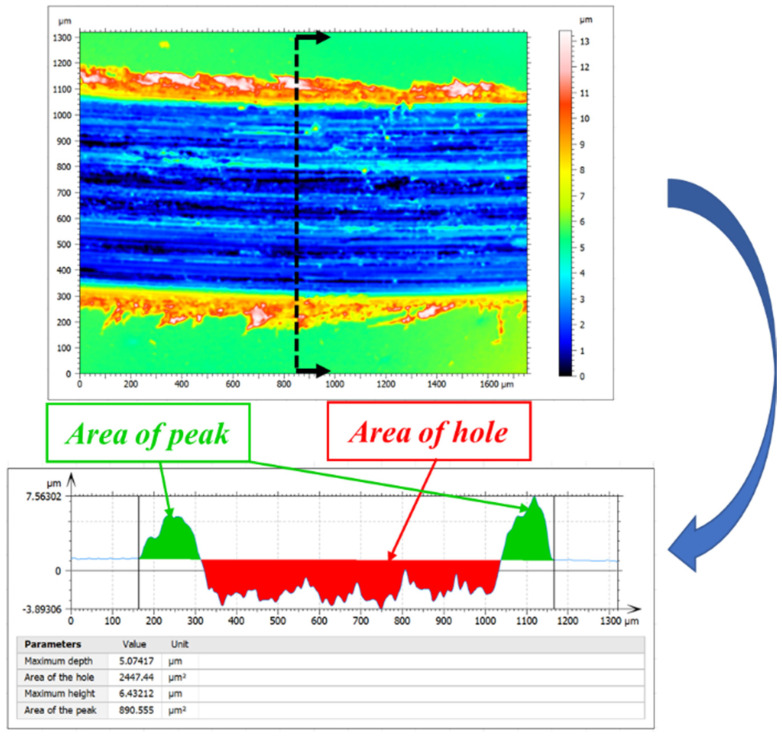
The cross-sectional morphology of the wear track obtained by 3D morphology acquisition (Sensofar Metrology) to calculate the wear rate.

**Figure 4 materials-14-02964-f004:**
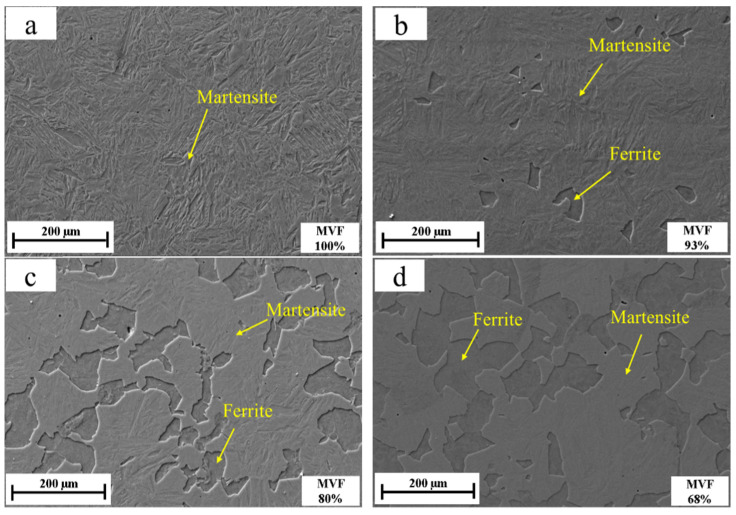
SEM images of the six generated microstructures under various intercritical holding times: (**a**) 0 min, (**b**) 1 min, (**c**) 2 min, (**d**) 3 min, (**e**) 6 min and (**f**) 7 min.

**Figure 5 materials-14-02964-f005:**
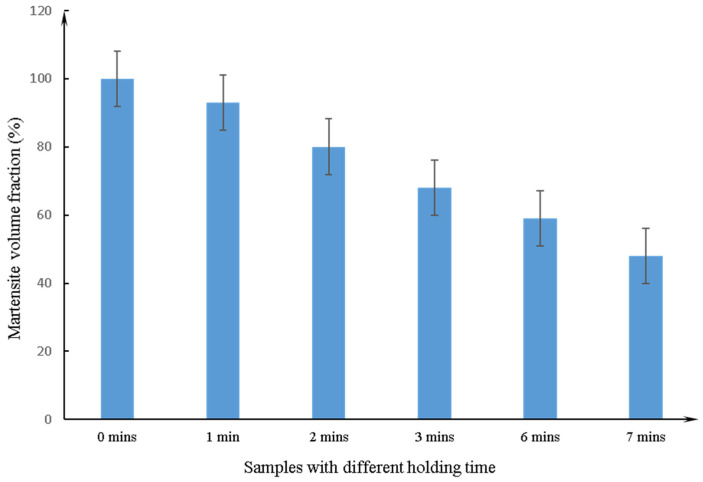
The MVF as a function of intercritical holding time at 725 °C.

**Figure 6 materials-14-02964-f006:**
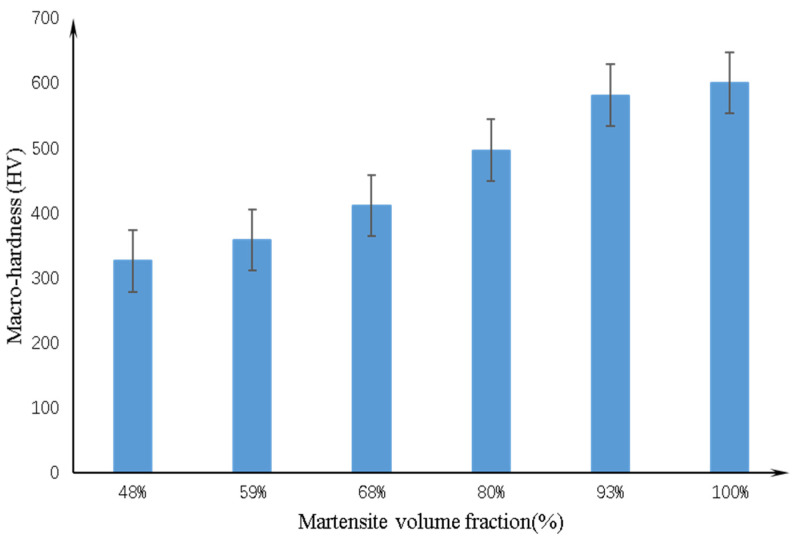
The variation of macrohardness as a function of MVF.

**Figure 7 materials-14-02964-f007:**
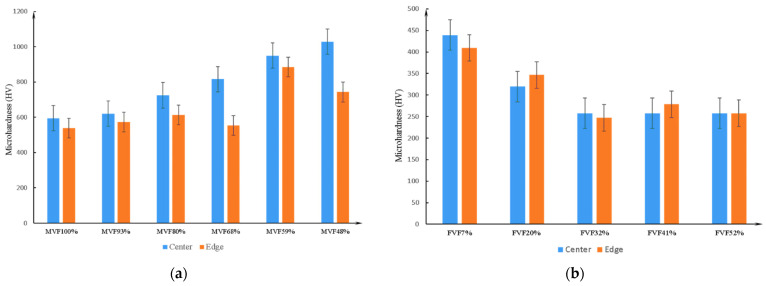
The microhardness changes with the volume fraction of: (**a**) martensite, (**b**) ferrite.

**Figure 8 materials-14-02964-f008:**
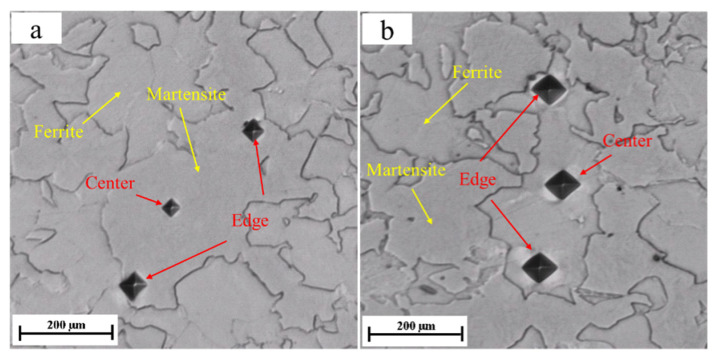
The changing of microhardness at different positions of different phases: (**a**) martensite, (**b**) ferrite.

**Figure 9 materials-14-02964-f009:**
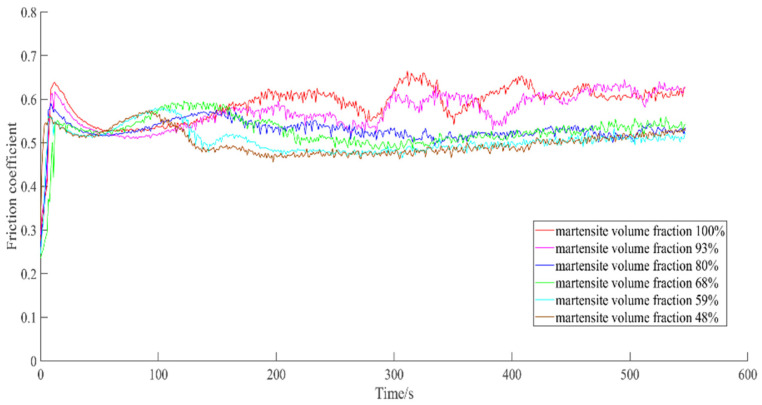
Variation of the friction coefficient with time under different MVFs.

**Figure 10 materials-14-02964-f010:**
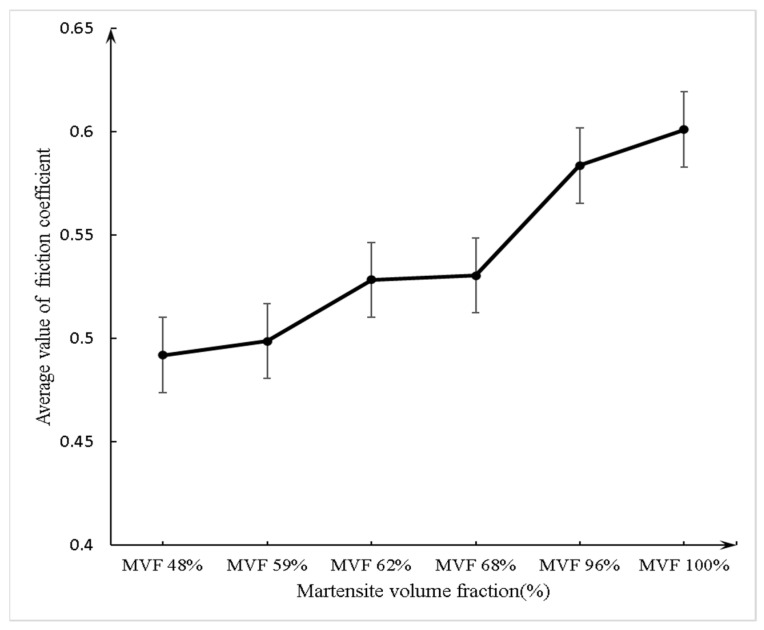
Variation of the average friction coefficient at a steady state with MVF.

**Figure 11 materials-14-02964-f011:**
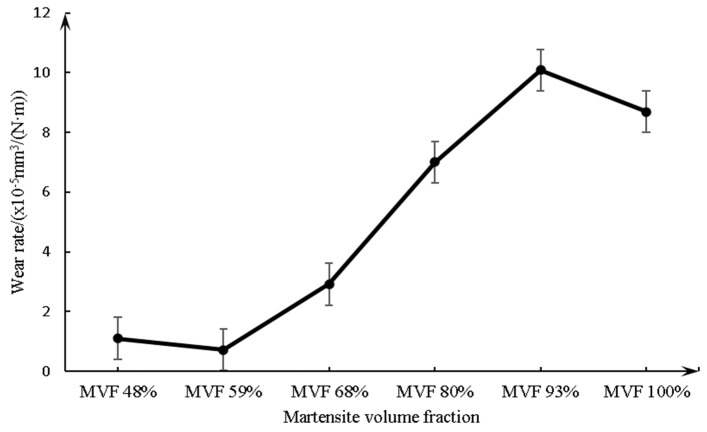
The wear rate as a function of MVF.

**Figure 12 materials-14-02964-f012:**
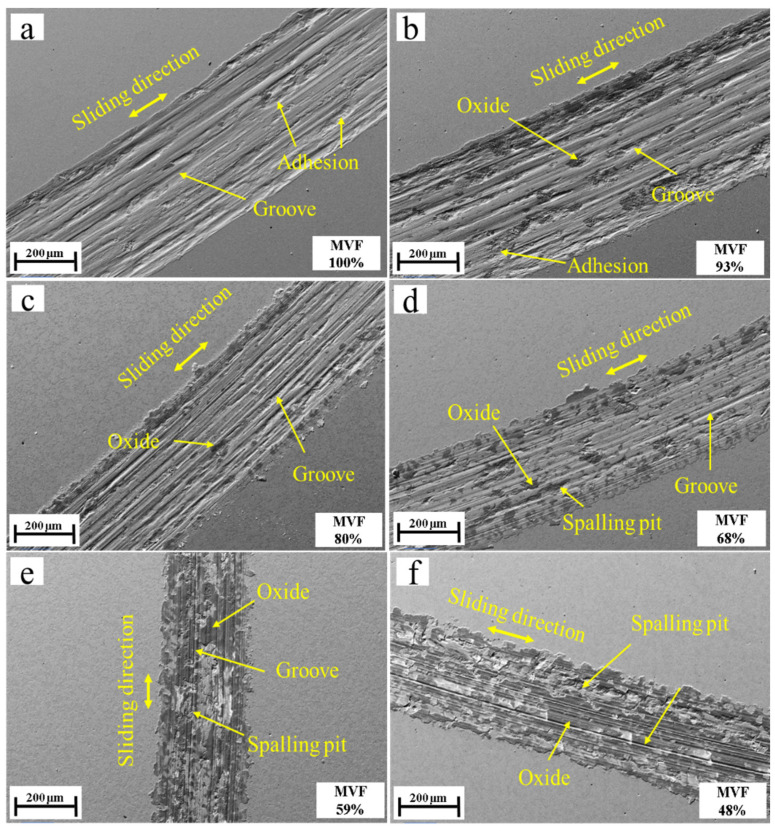
SEM images of wear tracks at MVF (**a**) 100%, (**b**) 93%, (**c**) 80%, (**d**) 68%, (**e**) 59% and (**f**) 48%.

**Figure 13 materials-14-02964-f013:**
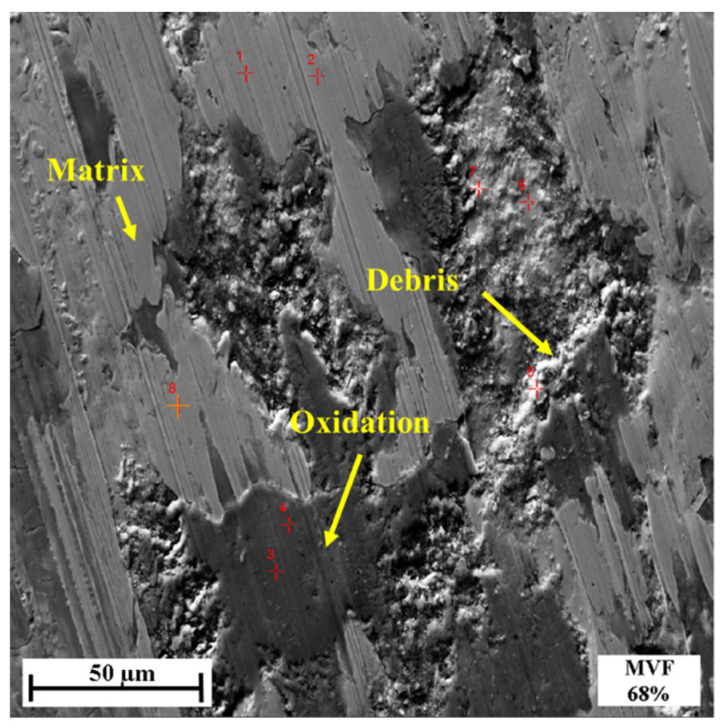
EDXS analysis of the points at MVF 68%.

**Figure 14 materials-14-02964-f014:**
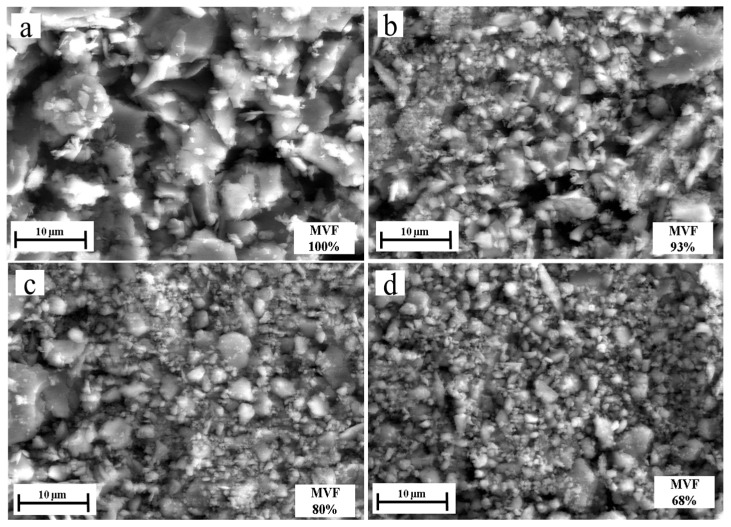
The size of debris as a function of MVF by SEM: (**a**) 100% martensite, (**b**) 93% martensite, (**c**) 80% martensite, (**d**) 68% martensite, (**e**) 59% martensite and (**f**) 48% martensite.

**Figure 15 materials-14-02964-f015:**
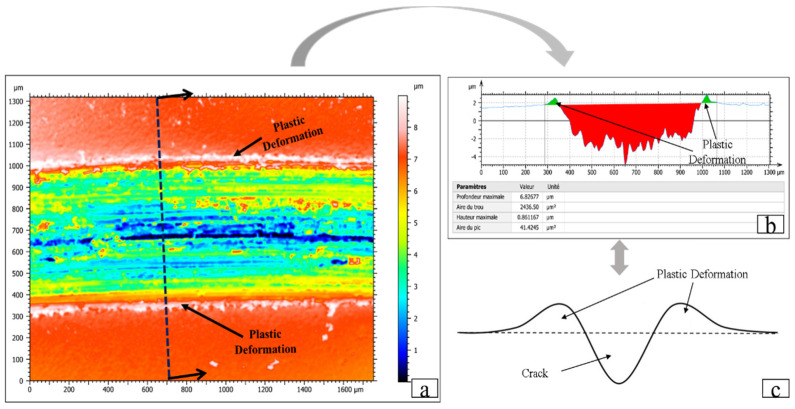
The plastic deformation acquired by SensoMAP: (**a**) the wear groove, (**b**) the cross-sectional view of the wear groove, (**c**) schematic drawing of the cross-sectional area of the wear groove and definition of the plastic deformation.

**Figure 16 materials-14-02964-f016:**
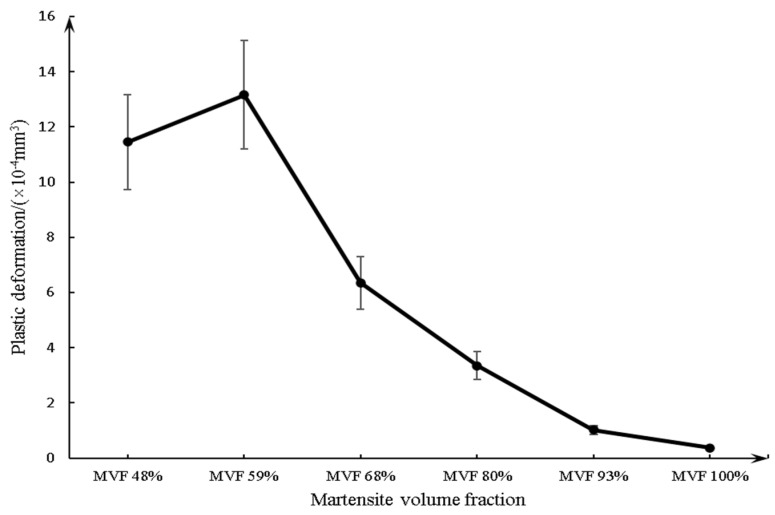
The plastic deformation as a function of MVF.

**Table 1 materials-14-02964-t001:** The chemical composition of 25CD4 steel (weight %).

C	Mn	Si	S	P	Cr	Ni	Mo
0.25	≤0.90	≤0.40	≤0.035	≤0.035	≤1.20	-	≤0.25

**Table 2 materials-14-02964-t002:** Oxygen concentration (in at%) of wear tracks at different MVFs.

MVF	Test 1 (%)	Test 2 (%)	Test 3 (%)	Average (%)
MVF100%	7.57	5.23	7.53	6.78
MVF93%	6.88	12.28	13.13	10.76
MVF80%	18.22	19.51	19.12	18.95
MVF68%	21.18	18.58	21.98	20.58
MVF59%	29.33	26.64	27.78	27.92
MVF48%	31.68	28.07	27.2	28.98

**Table 3 materials-14-02964-t003:** Elemental composition (in at%) of wear tracks and debris at MVF 59% by EDXS analysis.

Label	No	W% (C)	W% (O)	W% (Al)	W% (Si)	W% (Cr)	W% (Mn)	W% (Fe)	W% (Mo)
1	1	0.45	6.77	0.05	0.24	0.97	0.85	90.22	0.46
2	2	1.00	2.84	0.00	0.15	0.97	0.83	94.00	0.22
3	3	0.88	35.02	0.32	0.18	0.58	0.30	62.34	0.37
4	4	0.80	35.39	0.40	0.20	0.60	0.49	61.79	0.33
5	5	1.44	44.00	0.61	0.22	0.41	0.42	52.81	0.08
6	6	1.24	3.09	0.05	0.20	1.12	0.76	93.23	0.30
7	7	1.43	23.33	0.21	0.23	0.80	0.56	73.30	0.14
8	8	1.02	3.97	0.00	0.18	0.94	0.76	92.78	0.34

## Data Availability

Not applicable.

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
