# Peer review of "Effect of Martensite Volume Fraction on Oxidative and Adhesive Wear"

_materials, 2021, doi:10.3390/ma14112964_

Round 1
Reviewer 1 Report
The proposed article can be commented as follows:
Although the problem can be of interest for engineering practice, some things need to be improved.
1. Starting from the introduction onwards, there are many abbreviations that are not defined in a timely manner
2, .three main reason .... what about fatigue?
3. Chapter 2.1, this flat specimen, with a diameter of 55 mm .... confusion is created about the shape of the specimen.
4The paper is not clearly written. There are insufficiently defined things. Wear should be examined for precisely drined surface conditions and defined specimen temperatures
5. Books are always current, although older editions, but here there is a lot of literature of older years: references: 3/1974; 8/1978; 19/1981; 28/1956.
Author Response
Dear reviewer,
Thank you for your kind letter for Materials-1213385. We have revised the manuscript in accordance with every comment as you suggested, and carefully proofread the manuscript again to minimize typographical, grammatical, and bibliographical errors.
Our description on revision according to the comments as follows.
Comments and Suggestions for Authors
The proposed article can be commented as follows:
Although the problem can be of interest for engineering practice, some things need to be improved.
- Starting from the introduction onwards, there are many abbreviations that are not defined in a timely manner
The author’s response:
Thanks for your kind comment. The abbreviations have been explained in the introduction at 1st page 10th line and 2nd page 77th line, which are highlight in yellow color in the revised manuscript.
2.three main reason .... what about fatigue?
The author’s response:
Thanks for your kind comment. It has been revised on the 1st page 20th line according to your suggestions, which is highlight in yellow color in the revised manuscript.
- Chapter 2.1, this flat specimen, with a diameter of 55 mm .... confusion is created about the shape of the specimen.
The author’s response:
Thanks for your kind comment. It has been revised on the 2nd page 81st line according to your suggestions, which is highlight in yellow color in the revised manuscript.
The experimental samples have a diameter of 55 mm and a thickness of 3 mm, which were cut directly from the initial steel.
4.The paper is not clearly written. There are insufficiently defined things. Wear should be examined for precisely drined surface conditions and defined specimen temperatures
The author’s response:
Thanks for your kind comment. The content has been added to chapter 2.3 of the 3rd page 127th line according to your suggestions, which is highlight in yellow color in the revised manuscript.
Before the friction tests, each sample is cleaned for 10 minutes by ultrasound in ethanol, and then the surface is kept dry. All experiments are carried out at room temperature (23°C) in the air atmosphere, and the relative humidity is maintained at 50~60%.
- Books are always current, although older editions, but here there is a lot of literature of older years: references: 3/1974; 8/1978; 19/1981; 28/1956.
The author’s response:
Thanks for your kind comment. The related literatures have been replaced according to your suggestions, which is highlight yellow color at the chapter of reference.
Cui, X. H., et al., Wear characteristics and mechanisms of H13 steel with various tempered structures. Journal of materials engineering and performance, 2011. 20(6): p.1055-1062.
Zhang, Jiecen, et al. Effect of martensite morphology and volume fraction on strain hardening and fracture behavior of martensite–ferrite dual phase steel. Materials Science and Engineering: A, 2015.627: p.230-240.
Wang S Q, Wei M X, Wang F, et al. Effect of morphology of oxide scale on oxidation wear in hot working die steels[J]. Materials Science and Engineering: A, 2009, 505(1-2): p.20-26.
We greatly appreciate all helpful comments and suggestions in our manuscript, because they are valuable in improving the quality of our manuscript.
Please do not hesitate to contact us if you have any question. We are looking forward to hearing from you soon.

Reviewer 2 Report
This is an interesting paper which studies how the volume fraction of martensite affects the type and amount of wear in a dual phase steel.
The techniques employed in the work are numerous and allow the conclusions indicated. The measured values of the coefficient of friction, the wear, the composition, and shape of the tracks determined by EDX, and SEM were used.
I have a few minor comments:
Pag 2, line 5Pag 2, line 57 Capitalise trevisiol.
Pag 2, line 64 consider add an article
Pag 3, line 118 consider change "on" to "in".
All hardness values must be accompanied by HV not Hv.
Pag 6, line 181 Three citations are listed, consider changing "work has" to "works have".
Pag. 8, line 217 Delete "And"
Pag. 8, line 223 change "It" to "it"
Pag. 10, line 265 and pag. 11, line, 270 consider adding an article before "function"
Pag. 12, line 330 consider change "with" to "as"
Author Response
Dear reviewer,
Thank you for your kind letter for Materials-1213385. We have revised the manuscript in accordance with every comment as you suggested, and carefully proofread the manuscript again to minimize typographical, grammatical, and bibliographical errors.
Our description on revision according to the comments as follows.
Comments and Suggestions for Authors
This is an interesting paper which studies how the volume fraction of martensite affects the type and amount of wear in a dual phase steel.
The techniques employed in the work are numerous and allow the conclusions indicated. The measured values of the coefficient of friction, the wear, the composition, and shape of the tracks determined by EDX, and SEM were used.
I have a few minor comments:
1.Pag 2, line 5Pag 2, line 57 Capitalise trevisiol.
The author’s response:
Thanks for your kind comment. It has been revised at Pag. 2, line 58 according to your suggestions, which is highlight in yellow color in the revised manuscript.
2.Pag 2, line 64 consider add an article
The author’s response:
Thanks for your kind comment. It has been revised at Pag. 2, line 73 according to your suggestions, which is highlight in yellow color in the revised manuscript.
3.Pag 3, line 118 consider change "on" to "in".
The author’s response:
Thanks for your kind comment. It has been revised at Pag. 4, line 131 according to your suggestions, which is highlight in yellow color in the revised manuscript.
4.All hardness values must be accompanied by HV not Hv.
The author’s response:
Thanks for your kind comment. All unit of hardness has been changed from ‘Hv’ to ‘HV’ at Pag. 6, line 186 and at Pag. 6, line 188 according to your suggestions, which is highlight in yellow color in the revised manuscript.
5.Pag 6, line 181 Three citations are listed, consider changing "work has" to "works have".
The author’s response:
Thanks for your kind comment. It has been revised at Pag. 6, line 195 according to your suggestions, which is highlight in yellow color in the revised manuscript.
6.Pag. 8, line 217 Delete "And"
The author’s response:
Thanks for your kind comment. It has been deleted at Pag. 8, line 233 according to your suggestions, which is highlight in yellow color in the revised manuscript.
7.Pag. 8, line 223 change "It" to "it"
The author’s response:
Thanks for your kind comment. It has been revised at Pag. 8, line 239 according to your suggestions, which is highlight in yellow color in the revised manuscript.
8.Pag. 10, line 265 and pag. 11, line, 270 consider adding an article before "function"
The author’s response:
Thanks for your kind comment. It has been revised at Pag.10, line 287 and Pag.11, line 292 according to your suggestions, which is highlight in yellow color in the revised manuscript.
9.Pag. 12, line 330 consider change "with" to "as"
The author’s response:
Thanks for your kind comment. It has been revised at Pag. 12, line 352 according to your suggestions, which is highlight in yellow color in the revised manuscript.
We greatly appreciate all helpful comments and suggestions in our manuscript, because they are valuable in improving the quality of our manuscript.
Please do not hesitate to contact us if you have any question. We are looking forward to hearing from you soon.

Reviewer 3 Report
The paper investigates the effect of percentage of MVF in dual phase steel on friction coefficient, wear rates, oxidation, adhesion, plastic deformation and size of debris, after wear tests at a ball-on-disk tribometer.
I find this paper very interesting, well written and clearly presented.
A lot of sofisticated instruments were used to achieve the results, as SEM, EDX. 3D optical profilometry. Nevertheless I suggest some improvements and to clarify some contents.
- In the introduction, at rows 35-42 authors state that other researchers found that wear rate of dual phase steels decrease with increasing MVF. In this paper the results demonstrate an inverse trend, at least between 59% and 93% MVF. I wonder if you can explain better why you achieve this versus literature results.
- About equation 1: please specify the measuring units of Q. I see in figure 11 that the units should me mm3/m, so I do not understand the role of F in the equation. Please clarify.
- In figure 3 and figure 15 the numbers and labels are barely readable
- In section 2.3, when citing figure 3, please explain what peaks represent (plastic deformation), as it is not mentioned.
- Figure 15 is not mentioned throughout the text.
Author Response
Dear reviewer,
Thank you for your kind letter for Materials-1213385. We have revised the manuscript in accordance with every comment as you suggested, and carefully proofread the manuscript again to minimize typographical, grammatical, and bibliographical errors.
Our description on revision according to the comments as follows.
Comments and Suggestions for Authors
The paper investigates the effect of percentage of MVF in dual phase steel on friction coefficient, wear rates, oxidation, adhesion, plastic deformation and size of debris, after wear tests at a ball-on-disk tribometer.
I find this paper very interesting, well written and clearly presented.
A lot of sofisticated instruments were used to achieve the results, as SEM, EDX. 3D optical profilometry. Nevertheless I suggest some improvements and to clarify some contents.
1.In the introduction, at rows 35-42 authors state that other researchers found that wear rate of dual phase steels decrease with increasing MVF. In this paper the results demonstrate an inverse trend, at least between 59% and 93% MVF. I wonder if you can explain better why you achieve this versus literature results.
The author’s response:
Thanks for your kind comment. From a tribological point of view, each friction system may behave in different ways which will lead to different friction and wear results. The parameters affecting friction and wear mainly include contact conditions (ball-disk contact, pin-disk contact, etc.), movement mode (dry sliding, lubrication, fretting wear, etc.), load, speed, and so on. Due to the difference in the friction system, the main wear mechanism will be different. For the experiment of Trevisiol [1], the reciprocating linear dry sliding contact was used by cylindrical pin (DP steel) and abrasive paper (abrasive particle size 15~200um), and the load was 50~110N. Experimental results showed that under this friction system, the main wear mechanism is abrasive wear. For Saghafian [2], the experiment used a unidirectional rotary motion contact method with a cylindrical pin (AISI 52100) and a disc (DP steel), the load change is 61~82N, and the fixed sliding speed is 1.2m/s. The experimental results showed that the main wear mechanism is delamination wear.
Compared with the above experimental conditions, the experiment in this paper used a ball-disk (ball: Al2O3 ceramic; disc: DP steel) contact method, dry reciprocating sliding motion, a load of 15N, and an average sliding speed of 0.013m/s. According to SEM pictures and EDX analysis, the main wear mechanism in this experiment is oxidative wear.
It can be seen that due to different friction systems, the microstructure (volume fraction of martensite) has different effects on the friction coefficient and wear mechanism of dual-phase steel.
This can explain why the experimental results in this article are different from those in the literature.
2.About equation 1: please specify the measuring units of Q. I see in figure 11 that the units should me mm3/m, so I do not understand the role of F in the equation. Please clarify.
The author’s response:
Thanks for your kind comment. The unit of figure 11 has been changed to mm3/(N·m). I apologize for your doubts caused by my spelling error. The calculation of the wear rate used in this paper is shown in equation 1. Among them, (Shole-Speak)×L is the total wear volume, F means normal load (N) and D means the total sliding distance (m). The calculation principle of wear rate is based on Archard law: Wr=V/FL[3], where V is the wear volume (mm3), F is the load (N), and L is the sliding distance (m). Wear rate represents the material volume loss per unit sliding distance and unit load.
3.In figure 3 and figure 15 the numbers and labels are barely readable.
The author’s response:
Thanks for your kind comment. Figure 3 and figure 15 have been replaced.
4.In section 2.3, when citing figure 3, please explain what peaks represent (plastic deformation), as it is not mentioned.
The author’s response:
Thanks for your kind comment. It can be seen from the cross-sectional profile of the wear track (Fig.3) that the ‘peak’ refers to the amount of plastic deformation due to extrusion during the friction process, and the ‘hole’ refers to the wear loss of the disc material during the friction process. And the relevant text explanation has been added to the 4th page 143rd line, which is highlight in yellow color in the revised manuscript. Figure 15 was also changed showing the plastic deformation generated by friction
5.Figure 15 is not mentioned throughout the text.
The author’s response:
Thanks for your kind comment. The related description of Figure 15 has been added in the article, and it was highlighted on the 12nd page 316th line.
We greatly appreciate all helpful comments and suggestions in our manuscript, because they are valuable in improving the quality of our manuscript.
Please do not hesitate to contact us if you have any question. We are looking forward to hearing from you soon.
Reference:
- Trevisiol, C., A. Jourani, and S. Bouvier, Effect of martensite volume fraction and abrasive particles size on friction and wear behaviour of a low alloy steel. Tribology International, 2017. 113: p. 411-425.
- Saghafian, H. and S. Kheirandish, Correlating microstructural features with wear resistance of dual phase steel. Materials Letters, 2007. 61(14-15): p. 3059-3063.
- Li S, Wu X, Li X, et al. Wear characteristics of Mo-W-type hot-work steel at high temperature. Tribology Letters, 2016. 64(2): 1-12.

Reviewer 4 Report
The Authors investigated experimentally the effect of martensite volume fraction(MVF) on oxidative wear by using 25CD4 dual phase steel both from a microstructure point of view and from a tribological one. I have some concerns to be addressed point by point before consider this paper further.
Introduction
The introduction must focus on the scientific framework underlining in a clearer way the lack of knowledge in which the investigation is proposed. In particular from a tribological point of view the authors discuss the dual phase steel wear phenomena neglecting the entire tribosystems in which the results were found (coupling materials, test conditions). From a tribological point of view this is not a correct approach since each tribosystem could behaves in different ways depending on that. The authors should focus the discussion on similar cases (if any) proposed in their work, or declares or generalize the literature results in a logic way or, at least, declare the tribological conditions in which the results of other authors were found.
Methods
The tribological experiments were carried out by a ball/disc reciprocating tribometer (in this case should be better to name it ball on flat) with the aim mainly to measure the wear rate.
Usually these tests were performed, for repeatability and reproducibility purposes, under ASTM G133 standard. Did the Authors follow that standard? (test conditions, test procedures, data analysis……)
The authors declare “All of the specimens were tested at normal loads of 15 N and at a mean sliding velocity of 113 0.013 m/s” How the Authors calculated the mean sliding velocity? Did the author consider in their surface analysis the different sliding speed during the ball strokes?
Figures 3,7, 15, 16 are unclear and not readable. In my opinion, it is not scientific to report a capture screen of a commercial software as result.
The equations (2) and (3) should be referenced.
Regarding frictional tests, the authors wrote: “And the average friction 217 coefficient is the average value of the instantaneous friction coefficient in a steady state 218 (i.e. from 100 seconds to the end of the test)”…please give a look at DOI: 10.1109/I2MTC.2018.8409747…
Frictional results are not discussed in the framework of the proposed research, connecting it to the obtained surfaces measurements.
In the discussion of the results the authors claim “The oxide layer formed on 252 the worn track can reduce the wear, especially the adhesive wear, which is also observed 253 in this experiment.”. It is necessary to justify in details this!
Moreover, from row 314 to 321 of page 12 is reported a discussion to be clarified on the basis of the observed data.
In the conclusion the authors should underline main limitation of their investigation and few rows on future investigations about.
Author Response
Dear reviewer,
Thank you for your kind letter for Materials-1213385. We have revised the manuscript in accordance with every comment as you suggested, and carefully proofread the manuscript again to minimize typographical, grammatical, and bibliographical errors.
Our description on revision according to the comments as follows.
Comments and Suggestions for Authors
The Authors investigated experimentally the effect of martensite volume fraction(MVF) on oxidative wear by using 25CD4 dual phase steel both from a microstructure point of view and from a tribological one. I have some concerns to be addressed point by point before consider this paper further.
Introduction
The introduction must focus on the scientific framework underlining in a clearer way the lack of knowledge in which the investigation is proposed. In particular from a tribological point of view the authors discuss the dual phase steel wear phenomena neglecting the entire tribosystems in which the results were found (coupling materials, test conditions). From a tribological point of view this is not a correct approach since each tribosystem could behaves in different ways depending on that. The authors should focus the discussion on similar cases (if any) proposed in their work, or declares or generalize the literature results in a logic way or, at least, declare the tribological conditions in which the results of other authors were found.
The author’s response:
Thanks for your kind comment. The relevant content has been supplemented according to your suggestions. And the related tribological conditions of other authors were declared in the introduction at the 2rd page from 54th line to 72th line, which is highlight in yellow color in the revised manuscript.
steel, but there are huge differences in the experimental conditions of friction. The method adopted by Saghfian et al. [12] was a unidirectional sliding contact of the cylindrical steel pin and the steel disk, the load was 61~83N, and the sliding speed was relatively high (1.2m/s). The experiments of Trevisiol et al. [13] were focused on the contact of cylindrical pin samples against with the abrasive papers. The normal load was also relatively high (50~110N), and the sliding speed was constant at 0.06m/s. The test of Jha [9] was performed by a rubber wheel tribometer, in which the rubber wheel was directly rubbed against the flat steel sample in the vertical direction. The experimental load reached 48N and the sliding velocity was 3.27m/s. Due to the different tribological systems, the wear mechanism of dual phase steel is constantly changing. For example, for the frictional system of Trevisiol et al. [13], the main wear mechanism is abrasive wear; while in the tribological system of Saghafian et al. [12], the main wear mechanism is delamination wear. Considering that different tribological systems have a great effect on the properties of friction and wear, it is necessary to establish different frictional systems to investigate the relationship between their corresponding tribological behaviors and microstructures. Therefore, using the friction condition established in this paper to investigate the effect of the microstructure of dual phase steel on oxidative wear is a good supplement to the investigations of the tribological behavior of dual phase steel.
Methods
The tribological experiments were carried out by a ball/disc reciprocating tribometer (in this case should be better to name it ball on flat) with the aim mainly to measure the wear rate.
The author’s response:
Thanks for your kind comment. The related description has been replaced from ‘ball/disc’ to ‘ball on flat’ in the total article which is highlight in yellow color in the revised manuscript (75th line).
Usually these tests were performed, for repeatability and reproducibility purposes, under ASTM G133 standard. Did the Authors follow that standard? (test conditions, test procedures, data analysis……)
The author’s response:
Thanks for your kind comment. For repeatability and reproducibility purposes, the test conditions of this experiment are shown as fellow: room temperature(23℃); air environment; relative humidity of 50~60%; fixed sliding speed; constant load; the Al2O3 ceramic ball is directly purchased commercially; the surface roughness(Ra) of the dual phase steel is 0.25~0.35um and its geometric dimensions are fixed (diameter 55mm, thickness 3mm); reciprocating stroke length 10mm; According to Chapters 4.3, 4.5, and 6.5 of ASTM G133 -02, in line with relevant requirements.
Test procedures: Before each experiment on tribometer, the test sample is mechanically cleaned by ultrasound for 10 minutes in ethanol, and then made a hot air dry, which met the requirements of the experimental procedure in Chapters 8.2.1,8.2.2, and 8.5.1 of ASTM G133 -02.
Data analysis: Each friction experiment is carried out 3 times, and the average value of the 3 experiments is used to represent the final friction coefficient test result. For the measurement of wear rate, the area scan is performed by taking three fixed positions for each wear track, and the 3 cross-sectional profile is obtained from each obtained area scan. In summary, each sample has 3 wear tracks, each wear track has 3 acquisition positions, and each position has 3 interface contour acquisitions, which means that the final wear result under each experimental condition is an average of 27 measurements. Compared with the wear measurement conditions in Chapter 9.3 of ASTM G133-02, it meets the relevant requirements.
In my opinion, the above test condition, test procedure, and measurement results meet the relevant conditions of repeatability and reproducibility.
In addition, the experimental conditions and experimental procedures have been supplemented to chapter 2.3 at the 3rd page 127th line, which is highlight in yellow color in the revised manuscript.
The authors declare “All of the specimens were tested at normal loads of 15 N and at a mean sliding velocity of 0.013 m/s” How the Authors calculated the mean sliding velocity? Did the author consider in their surface analysis the different sliding speed during the ball strokes?
The author’s response:
Thanks for your kind comment. The mean sliding velocity was calculated by the total sliding distances (7 meters) divided by the total experimental time (546 seconds).
The main objective of this article is to study the impact of martensite volume fraction on tribological behavior of dual phase steel. Another study is underway to study the effect of speed on the friction and wear of this steel.
Figures 3,7, 15, 16 are unclear and not readable. In my opinion, it is not scientific to report a capture screen of a commercial software as result.
The author’s response:
Thanks for your kind comment. According to your suggestions, all Figures which you mentioned have been replaced in the revised manuscript.
The equations (2) and (3) should be referenced.
The author’s response:
Thanks for your kind comment. These equations (2 and 3) have been referenced in the revised manuscript.
Regarding frictional tests, the authors wrote: “And the average friction 217 coefficient is the average value of the instantaneous friction coefficient in a steady state 218 (i.e. from 100 seconds to the end of the test)”…please give a look at DOI: 10.1109/I2MTC.2018.8409747…
The author’s response:
Thanks for your kind comment. Based on your suggestion, I apologize to confuse the concept of 'average friction coefficient' and 'the average value of friction coefficient in a steady state'. The experimental data used in this article is the average value of the friction coefficient at steady state, that is, the average value of the friction coefficient obtained from the experiment from 100s to 546s. The related phrase 'average friction coefficient' in this paper has been changed to provide a more accurate description. The corresponding title name of Figure 10 and the ordinate name have been corrected. The correction of the content has been highlighted at 8th page 233rd line in the revised manuscript.
Figure 9 shows the variation of friction coefficient with MVF. The average value of friction coefficient at steady state is acquired and calculated by the experimental data from 100 to 546 seconds. Figure 10 shows the average value of friction coefficient at steady state as a function of MVF and it remains between 0.49~0.60
Frictional results are not discussed in the framework of the proposed research, connecting it to the obtained surfaces measurements.
The author’s response:
Thanks for your kind comment. The relevant explanation has been supplemented at the 11st page 302nd line, which is highlight in yellow color in the revised manuscript.
Compared with adhesive wear, the compacted oxide layer can effectively reduce the real contact area and junctions, which may bring down the coefficient of friction. The results of EDX analysis show that the oxidative wear increases with the decreasing MVF, which explains the phenomenon that friction coefficient decreases with the decreasing MVF.
In the discussion of the results the authors claim “The oxide layer formed on the worn track can reduce the wear, especially the adhesive wear, which is also observed in this experiment.”. It is necessary to justify in details this!
The author’s response:
Thanks for your kind comment. The relevant explanation has been supplemented at the 9th page 266th line, which is highlight in yellow color in the revised manuscript.
From the EDX results and SEM micrograph analysis, a great number of tribo-oxides appeared on worn surfaces of dual phase steel with the decreasing MVF. The morphology of worn surfaces gradually presented smooth undelaminated regions which were typical track of oxidative wear. Therefore, the previously mentioned compression film formed by wear debris can be definitely determined as an oxide film. Due to the existence of the oxide film, it effectively avoids direct contact between the frictional matrix. As the friction continues, the debris of oxides that remained in the wear track is better compacted on the contact surface. This also promotes the formation and maintenance of the oxide layer. Since the existence of the hard oxide layer, the wear rate gradually decreases, especially the adhesive wear, which is also observed in this experiment.
Moreover, from row 314 to 321 of page 12 is reported a discussion to be clarified on the basis of the observed data.
The author’s response:
Thanks for your kind comment. The relevant description has been supplemented at the 13rd page from 325th line to 345th line, which is highlight in yellow color in the revised manuscript.
In the conclusion the authors should underline main limitation of their investigation and few rows on future investigations about.
The author’s response:
Thanks for your kind comment. The related description has been added at 13rd page 359th line, which is highlight in yellow color in the revised manuscript.
During the sliding process, adhesion and junctions occur only on the real contact areas, where extremely high contact pressure causes the plastic deformation of the matrix. Compared with other regions, there is a higher temperature at plastically-deformed contacting areas. Thus, these plastically deformed areas are preferential locations for oxidation [38, 39]. As the plastic deformation of the substrate increases, more and more cracks would be formed and propagated [40, 41]. This could easily lead to the oxygen enter these cracks and react with steel substrate to form oxides inside the matrix. It is obvious that a large plastic deformation would occur in a softer matrix under the same normal load and sliding speed so that producing more tribo-oxides. Due to the low normal load and sliding velocity, the plastic deformation generated during wear is helpful to retain more tribo-oxides. The model proposed by Quinn [42, 43] can also be concluded that oxidative mild wear usually generates under low normal load and velocity, which wear property is only related with tribo-oxides. As the low-speed friction proceeds, the oxide remaining in the wear track is compacted into a hard tribo-oxide layer to prevent metal-metal contact and thus reduce wear.
According to the results of macro-hardness test (Figure 6) and the plastic deformation measurement (Figure 16), it can be obtained that the macro-hardness of the dual phase steel matrix decreases with the decreasing MVF, the plastic deformation increases with the decreasing MVF. Meanwhile, owing to the oxide increases with plastic deformation, oxidative wear increases with decreasing MVF. This reasonably explains the relationship between the microstructure of dual phase steel and oxidative wear.
We greatly appreciate all helpful comments and suggestions in our manuscript, because they are valuable in improving the quality of our manuscript.
Please do not hesitate to contact us if you have any question. We are looking forward to hearing from you soon.

Round 2
Reviewer 1 Report
It would be useful to improve the quality of the paper, in general, in many chapters.
However, if it is not to do so easy, it is not need to make any trouble, since some things are improved.
Reviewer 4 Report
The Authors addressed in satisfactory way my concerns.